# Enhanced Metabolome Coverage and Evaluation of Matrix Effects by the Use of Experimental-Condition-Matched ^13^C-Labeled Biological Samples in Isotope-Assisted LC-HRMS Metabolomics

**DOI:** 10.3390/metabo10110434

**Published:** 2020-10-27

**Authors:** Asja Ćeranić, Christoph Bueschl, Maria Doppler, Alexandra Parich, Kangkang Xu, Marc Lemmens, Hermann Buerstmayr, Rainer Schuhmacher

**Affiliations:** 1Department of Agrobiotechnology, Institute of Bioanalytics and Agro-Metabolomics, IFA-Tulln, University of Natural Resources and Life Sciences Vienna (BOKU), Konrad-Lorenz-Strasse 20, 3430 Tulln an der Donau, Upper Austria, Austria; asja.ceranic@boku.ac.at (A.Ć.); christoph.bueschl@boku.ac.at (C.B.); maria.doppler@boku.ac.at (M.D.); alexandra.parich@boku.ac.at (A.P.); kangkang.xu@boku.ac.at (K.X.); 2Department of Agrobiotechnology, Institute of Biotechnology in Plant Production, IFA-Tulln, University of Natural Resources and Life Sciences Vienna (BOKU), Konrad-Lorenz-Strasse 20, 3430 Tulln an der Donau, Upper Austria, Austria; marc.lemmens@boku.ac.at (M.L.); hermann.buerstmayr@boku.ac.at (H.B.)

**Keywords:** untargeted metabolomics, internal standard, deoxynivalenol, abiotic stress of wheat, matrix effects, GLMe-IS

## Abstract

Stable isotope-assisted approaches can improve untargeted liquid chromatography-high resolution mass spectrometry (LC-HRMS) metabolomics studies. Here, we demonstrate at the example of chemically stressed wheat that metabolome-wide internal standardization by globally ^13^C-labeled metabolite extract (GLMe-IS) of experimental-condition-matched biological samples can help to improve the detection of treatment-relevant metabolites and can aid in the post-acquisition assessment of putative matrix effects in samples obtained upon different treatments. For this, native extracts of toxin- and mock-treated (control) wheat ears were standardized by the addition of uniformly ^13^C-labeled wheat ear extracts that were cultivated under similar experimental conditions (toxin-treatment and control) and measured with LC-HRMS. The results show that 996 wheat-derived metabolites were detected with the non-condition-matched ^13^C-labeled metabolite extract, while another 68 were only covered by the experimental-condition-matched GLMe-IS. Additional testing is performed with the assumption that GLMe-IS enables compensation for matrix effects. Although on average no severe matrix differences between both experimental conditions were found, individual metabolites may be affected as is demonstrated by wrong decisions with respect to the classification of significantly altered metabolites. When GLMe-IS was applied to compensate for matrix effects, 272 metabolites showed significantly altered levels between treated and control samples, 42 of which would not have been classified as such without GLMe-IS.

## 1. Introduction

Alongside with nuclear magnetic resonance (NMR) and gas chromatography-mass spectrometry (GC-MS), the most dominant technique in metabolomics studies is liquid chromatography-high resolution mass spectrometry (LC-HRMS). It facilitates high throughput measurements with high sensitivity and selectivity and can also be tailored for many different applications as well as chemical compound classes. Typically, metabolomics is employed to study metabolic alterations in biological extracts obtained from different experimental conditions (i.e., treatments and controls). While targeted approaches are focused on the evaluation of known metabolites from a predefined list, untargeted workflows aim at evaluation of the global detectable metabolome and deal with hundreds to thousands of metabolites that are mainly unknown. Therefore, statistically based data evaluation is essential in untargeted metabolomics in order to maintain high throughput also in all post-measurement evaluation steps. A typical data set of an untargeted metabolomics study that consists of control and treatment samples may be statistically evaluated by a comparison of their metabolic profiles (i.e., type and abundances of the detected compounds) that actually reflects the metabolic response to the treatment(s). For a reliable comparison and meaningful statistical analysis, the detected compounds should truly be metabolites of the studied biological system [1]; the detector response has to be approximately linear [2], and ideally, the extent of matrix effects between treatment and control should be similar to avoid a large portion of technical bias and to minimize the variance [3].

With respect to the origin of the detected signals and metabolites, the measured chromatograms contain not only compounds of biological origin but also many contaminants such as extraction solvents and plasticizers. Such contaminant signals may be reliably filtered out using stable isotope-assisted (SIA) approaches. In these, highly enriched global stable isotopically labeled biological extracts can be used for metabolome-wide internal standardization of native biological extracts under investigation. Together, the native and labeled metabolite features that originate from the biological sample will show highly characteristic artificial isotopolog patterns in the obtained mass spectra, which can be efficiently detected as biology-derived metabolic features with automated software (e.g., MS-DIAL, geoRge, HiTIME, MetExtract II, or X^13^CMS [4,5,6,7,8]). Moreover, the efficient filtering of all nonbiological signals and nonspecific metabolic features allows the convolution of all features originating from the same metabolite into so-called feature groups based on similarity of chromatographic peak shapes and retention times [1,8]. Thus, SIA approaches like MetExtract II allow the reliable detection of truly biology-derived metabolites. As many wheat metabolites like nitrogen containing compounds, sugars, organic acids, or phenolics can be newly induced upon stress [9]), SIA-LC-HRMS approaches may benefit from closely “experimental-condition-matched” globally labeled metabolite extracts as internal standards (GLMe-IS) rather than an unspecific reference metabolome in order to effectively detect newly stress-induced metabolites. Such extracts are obtained from ^13^C-labeled biological samples or organisms that are generated under similar experimental conditions as non-labeled from the actual biological experiment, ideally with the only difference being the isotopic enrichment of the applied labeled nutrients. Besides plants, also microbes like bacteria and fungi as well as mammalian cell cultures and tissues were successfully labeled [10,11].

Another factor that can potentially limit the reliability, accuracy, and interpretation of LC-HRMS data in comparative metabolomics studies are matrix effects. They occur in the ionization source of the mass spectrometer, in particular with electrospray ionization, when co-eluting metabolites alter the ionization efficiency of the analyte(s). As a consequence, either an increase or decrease in the ionization efficiency may occur that results in falsely higher or lower detected metabolite abundances compared to single compounds in pure solvents. While the exact mechanism of matrix effects remains unclear, it is generally assumed to occur as a result of competition between co-eluting metabolites for charge transfer and access to the surface of analyte containing eluent droplets in the ionization source of the mass spectrometer [12]. Along with coeluting compounds, ion pairing reagents in chromatography (i.e., HPLC additives) as well as buffering reagents may also affect ionization of the analytes. While ion pairing reagents are used to improve metabolite separation and peak shape during chromatography, buffers facilitate ion transmission during the ionization process in the mass spectrometer. Ion pairing reagents such as trifluoroacetic, pentafluoropropionic, and heptafluorobutyric acids are known to suppress ionization in negative mode, while tetraalkyl ammonium hydroxide and salts do so in positive mode [13]. Further, inorganic nonvolatile buffers such as phosphate and sulphate can cause salt deposits on the needle surface and in such a way reduce conductivity in the ionization source [14]. Therefore, when a mass spectrometer is used as the detector, ion pairing reagents should rather not be used. Instead, organic volatile buffers such as ammonium formate or ammonium acetate are rather recommended for buffering [13].

In untargeted metabolomics applications, matrix effects represent a challenge as each co-eluting metabolite is either considered as the analyte while the others can be seen as the matrix and vice versa. Therefore, all low molecular weight metabolites in the analyzed extract are of interest and some of those that cause significant matrix effects cannot simply be removed from the system by clean up or other types of sample preparation. Consequently, each metabolite may be affected by or contribute to alterations in ionization efficiency, the extent of which can change for the very same metabolite depending on the qualitative and quantitative compositions of the co-eluting metabolites [3,15]. This may represent an additional challenge for comparative studies that deal with biological samples of different conditions and thus metabolic profiles among the compared samples. Consequently, this may result in inaccurate assignment of metabolite abundance and eventually wrong judgment about differences in the metabolic composition of samples/organisms under investigation.

Unfortunately, it is currently impossible to quantify the matrix effects exactly [16,17], but at least for targeted approaches, there are different ways to estimate the extent of or account for ion suppression/enhancement such as the comparison between solvent and matrix calibration or injection of a matrix sample and post-column infusion of standard compounds into the column effluent as has been reviewed elsewhere [18,19]. For the reason explained above, it is however very difficult or even impossible to assess the existence or extent of matrix effects in the case of untargeted approaches. The best method to take into account for matrix effects is the addition of labeled metabolic constituents in the form of an internal standard to the native biological extract [3]. Potential differences in matrix effects between the biological samples under investigation can be corrected by using abundance ratios of native and labeled forms (normalized abundances) as a relative quantitative measure for each metabolite in the comparative study. Based on the assumption that native and the corresponding labeled metabolites always co-elute and experience ion suppression/enhancement to the same extent, the abundance ratio of the two is not affected as long as every native sample contains the same amount of labeled internal standard. In untargeted metabolomics studies that deal with hundreds to thousands of metabolites at the same time, it can be challenging to find a representative labeled internal standard. If only one or several labeled metabolites are used to internally standardize the entire investigated biological extract [20], most of the matrix effects could not be compensated for, since they can vary within seconds during analysis and strongly depend on the physicochemical properties of the (unknown) sample constituents [17]. Consequently, globally labeled ^13^C (or ^15^N or ^34^S)-labeled biological samples are best suited as internal standards since, here, the ratio of native and labeled forms of every metabolite that is shared between labeled and native samples is considered for compensation of ion suppression/enhancement [11].

The application of native and labeled sample mixtures in LC-HRMS-based untargeted metabolomics is implemented in many laboratories and is described as an effective method for detecting biology-derived metabolites and to improve (absolute, relative, or differential) quantification in metabolomics studies [5,21,22,23,24,25,26,27,28,29,30,31,32,33,34,35,36]. To exemplify a few alternative approaches, the MIRACLE (Mass isotopomer ratio analysis of U ^13^C-labeled extracts) approach, IROA (isotope ratio outlier analysis), and feature credentialing shall briefly be mentioned. The MIRACLE approach (Mass isotopomer ratio analysis of U ^13^C-labeled extracts) describes a quantification approach at the example of two yeast cultures [21] after normalization of native metabolite abundances relative to its ^13^C equivalent from ^12^C-^13^C mixtures. The IROA (isotope ratio outlier analysis) is similar to the MetExtract method and also employs ^12^C-^13^C mixtures for semiquantitative and qualitative analysis [22]. In contrast to the approach described in the present paper, a 95% isotopic enrichment is used by IROA to promote the reliable recognition of small molecules with a low number of carbon atoms per formula unit. Similarly, an extended ^12^C-^13^C approach was published under the name feature credentialing platform [23]. There, ^12^C-^13^C labeled samples are additionally mixed in a ratio of 1:2 (cells/cells), which enables an isotopic signature, which is specific enough to further increase the discrimination of biology-derived metabolite signals from those of contaminants.

Here, we aim to assess the performance of globally labeled biological samples for metabolome-wide and experimental-condition-matched (ECM) internal standardization. At the example of the comparative study of wheat, abiotic stress (i.e., toxin treatment) was used to show the benefits of experimental-condition-matched ^13^C-GLMe-IS for the global detection of (also treatment-specific) biology-derived metabolites as well as for the fast assessment of matrix effects as a measure of the accuracy in comparative quantification and judgement of significance.

## 2. Results and Discussion

The experimental-condition-matched GLMe-IS has potential to improve untargeted metabolomics approaches by increasing the confidence level for the detection of truly biology-derived metabolites and by compensating putatively different matrix effects between samples obtained upon different treatments.

To test this, metabolic profiles of toxin-stressed and mock (H_2_O)-treated control wheat ears in native and ^13^C-labeled form were analyzed with a well-established SIA untargeted metabolomics workflow [1,8].

The experimental setup used to generate the analyzed data set is shown in Figure 1. Wheat (*Triticum durum*) has either been grown in a native CO_2_ atmosphere (98.9% ^12^C) in the glasshouse or in a ^13^C-enriched atmosphere (99% ^13^C) in a customized cultivation chamber [37] to generate native and uniformly ^13^C-labeled (L) wheat, respectively. Both were either treated with water (control and L-control) or toxin (toxin and L-toxin). Extracts of labeled wheat ears, L-toxin and L-control, were mixed 1+1 (v/v) to obtain the experimental-condition-matched GLMe-IS that was subsequently spiked to each of the native sample extracts. By this, two sample types were obtained, which are termed toxin//GLMe-IS and control//GLMe-IS (see Figure 1a).

### 2.1. Comparison of the Metabolic Composition between Native Control and Toxin Samples

In order to compare the qualitative metabolic composition of toxin-stressed with control wheat ear samples, a list of metabolic features (characterized by native *m/z*, ^13^C *m/z*, retention time (RT), and abundances) was generated after processing the analyzed samples with MetExtract II [8]. According to the SIA approach, each metabolic feature in the obtained list is present in a native form as well as in a labeled form (originating from GLMe-IS) in at least one sample type (see the isotope pattern in Figure 1b). Metabolic features of the same metabolite were grouped into one feature group, and the most abundant ion (native metabolic feature) in the group was used to represent the respective metabolite.

#### 2.1.1. Qualitative Comparison of Control and Toxin Treatment

In total, 1080 putative metabolites were detected, for which the distribution of retention times and *m/z* are shown in form of a feature plot in Figure 2a; 990 metabolites were shared between control and toxin treatment, while 62 metabolites were detected exclusively in the toxin samples and 6 were detected in the control samples (Figure 2b). The remaining 22 metabolites were inconclusively detected (only in a fraction of the samples of the toxin or control groups) and are thus not counted towards either of the two groups.

A detailed annotation and identification of the detected metabolites was not the purpose of this study. However, to illustrate the biological relevance, the additionally detected treatment-specific metabolites shall briefly be summarized here. More than one-third of the 62 putative stress-specific compounds (toxin-stressed additional metabolites) are secondary metabolites, most of which were also found in previous studies [38,39], using the same LC-HRMS method. They largely derive from the aromatic amino acids phenylalanine tryptophan and tyrosine [38,39,40]. For example, feruloyltryptamine is one of the detected toxin-treatment-specific metabolites, which was identified (identification with confidence level 1 based on the scoring system from [41]). Feruloyltryptamine belongs to hydroxycinnamic acid amides that are well-characterized defense-related compounds in wheat from the phenylpropanoid pathway, which are produced by the plant for cell-wall strengthening [42] to prevent the plant pathogenic *Fusarium* from spreading beyond the initially infected wheat spikelet [42,43,44].

These treatment-specific and biologically presumably important metabolites were only detected with the application of experimental-condition-matched GLMe-IS. If any other labeled reference matrix would be used, e.g., ^13^C-labeled control samples or commercially available ^13^C-labeled wheat ears which are not ECM, these features of treatment-specific metabolites could not have been detected with the SIA-LC-HRMS approach. Although features of treatment-specific metabolites might in principle also be detected without SIA application, they would still largely be hidden among numerous non-biology-derived compounds, as has been demonstrated earlier [1]. Furthermore, the presented approach supplements metabolomics studies as it facilitates annotation and structure elucidation of detected metabolic features by taking advantage of the specific isotope pattern of native and labeled compound analogs from MS and MS/MS spectra (for more details, see also [39]).

#### 2.1.2. Comparative Quantification of Control and Toxin Samples

The differential quantitative analysis showed that toxin treatment has a significant effect on the profiles of detected metabolites of the wheat extracts. This includes both treatment- and control-specific as well as relative abundances of shared metabolites, as shown in Figure 3. Both sample types are separated well along the first principal component, which covers 61% of the total variance between the two experimental groups (Figure 3a). A univariate comparison revealed that 282 (26%) of all native metabolites (common for control and toxin) were significantly different with a fold-change of at least 2 or 0.5 and a *p*-value below 0.05 (Figure 3b). From these, 89 native metabolites were significantly more abundant in control samples while the majority (*n* = 193) showed higher levels in toxin samples.

To fully rely on the measured abundances of the native metabolites and to draw a meaningful conclusion from a typical metabolomics study, ion suppression/enhancement (i.e., matrix effects) have to be evaluated. While ion suppression cannot be fully avoided in complex biological extracts, they should at least be similar for the samples derived upon application of different experimental conditions. However, the matrix effects, which derive from competition for charges in the ion source of the mass spectrometer, vary with the physicochemical properties of co-eluting metabolites (i.e., proton affinity and surface activity [13]) and might therefore be stronger or weaker, depending on the metabolic composition of the control- and toxin-treated samples. Thus, in the presented study, in which the two sample types have a different metabolic composition with respect to the detected native metabolites as well as different abundances of metabolites shared between the two experimental treatments (shown in Figure 3b), comparative quantification may be inaccurate if the matrix effects are not properly accounted for.

### 2.2. Investigation of Putative Matrix Effects and Applicability of the Data in Comparative Metabolome Studies

In a typical untargeted metabolomics experiment, matrix effects remain hidden as all biochemical small molecules constitute the matrix and at the same time are the objects under investigation. However, matrix effects can be investigated by the use of globally ^13^C-labeled material for internal standardization. To this end, absolute and normalized (internal standardized) abundances of the native signals are compared in different ways.

#### 2.2.1. Improvement of Quantification by Internal Standardization

Native metabolite abundances and ratios of native to labeled (contained in the pooled ^13^C extract (GLMe-IS)) were used to test whether comparative quantification of native ^12^C metabolites can be improved by internal standardization.

To assess quantification on a single metabolite level, first, the fold-change values (toxin/control) were calculated after normalization of ^12^C peak areas (A_Control_ and A_Toxin_) to ^13^C peak areas (A_GLMe-IS_) and compared to the fold-changes derived from absolute ^12^C peak areas. For this, the ratio of the two-fold-change values of the ^12^C peak areas (with and without internal standardization) were plotted against (A_Toxin_/A_GLMe-IS_)/(A_Control_/A_GLMe-IS_) (Figure 4a). Values close to 100% on the y-axis indicate that the obtained fold-change values before and after internal standardization are similar, and therefore, the data suggest that the two experimental sample types control and toxin (i.e., the two matrices) produce similar matrix effects on average. The relatively low discrepancy between the fold-change values with and without normalization can be explained by the high extent of similarity between the matrices tested here. Only 6% of the detected metabolites were exclusively found in one of the sample types. In other cases, where sample composition differs much more, it has been shown that the ionization efficiency of many investigated metabolites were affected, especially when the same metabolites were compared in different biofluid matrices such as urine, saliva, blood plasma, serum, etc. [3,45]. In case of such largely differing matrices, normalization of abundances to those of the labeled internal standards enabled compensation of differences in ionization efficiency of the investigated metabolites [45,46,47]. Compensation of ion suppression by GLMe-IS enables reliable comparison of the metabolic composition of largely differing wheat tissues like root, stem, leaf, spikelet, rachis, etc. [48,49]. It shall be mentioned that this type of reliability assessment in comparative quantification can only be accomplished by the application of experimental-condition-matched GLMe-IS, as all detected biology-derived metabolites (including treatment-specific) are present in both a native and ^13^C-labeled form in the investigated samples.

#### 2.2.2. Comparative Statistical Analysis of Native Metabolome Abundances in Toxin and Control Performed with Absolute and Normalized Abundances

While on average matrix effects were similar in both sample types (Figure 4a), another aspect that also has to be demonstrated to rely on the comparative quantification based on ^12^C feature abundances (as is the case for a typical state-of-the-art metabolomics study) is to critically evaluate the outcome of significance testing. Since the purpose of a typical untargeted metabolomics study is to find metabolites that are significantly altered by the treatment, it is crucial to verify that the comparison of toxin-treated and control samples leads to similar results when absolute metabolite abundances compared to abundance ratios obtained after normalization are used. In this context, normalized abundances (i.e., matrix effect corrected) can be considered as a reference for the comparison. Any deviation in the classification of significance with and without internal standardization (i.e., absolute abundances versus normalized abundances) may indicate a bias due to matrix effects.

To test this, metabolites in which the abundance was significantly altered under stress were evaluated by the use of ^12^C peak areas with and without internal standardization (Figure 4b). In total, 230 metabolites are reported to be significantly different between toxin and control samples by both approaches. Forty-two of those were only reported to be significantly different when internally standardized, and thus, normalized abundances were employed while another 52 metabolites were exclusively with absolute abundances. This clearly indicates that, if matrix effects are not corrected, around 19% of truly significantly differing metabolites (i.e., the one obtained with normalized abundances) would be falsely excluded from this group.

It can be seen from the feature map (Appendix A, Appendix A) that these metabolites are distributed throughout the whole chromatogram. This indicates that metabolites of different polarities and *m/z* range may be affected. Further, Figure 4a shows that the majority of these metabolites are distributed around the mean fold-change ratio threshold (2 or 0.5), while only a few showed a mean fold-change ratio that is higher than 2 or lower than 0.5.

In conclusion, our results demonstrate that, if abundances of detected native metabolites are not corrected for matrix effects, the reliability or correctness of the statistically assisted assessment of significantly different metabolites between two experimental conditions can be affected. In order to really prove that internal standardization of native peak areas by ^13^C areas is not biased and thus significance testing is reliable, absolute matrix effects would have to be estimated by comparison with authentic standards in pure solvent followed by significance testing. Since this is not feasible in current untargeted applications, internal standardization with GLMe-IS is considered the best choice at present.

#### 2.2.3. Evaluation of Matrix Effects on the Abundances of Labeled Metabolite Signals

GLMe-IS does also offer another way to test for matrix effects. For the following section, a statistical analysis was carried out that compared the peak abundances of labeled metabolites in the two sample types toxin//GLMe-IS and control//GLMe-IS. Both contain the same aliquot of the pooled GLMe-IS but differ in the type of native biological extracts. Therefore, the native metabolome can be regarded as the “matrix” while the ^13^C-labeled biochemical constituents can be seen as the metabolites of interest that are the same in both sample types with respect to identity and abundance. If the two native matrices (i.e., control or toxin) do not influence the abundance of the ^13^C labeled metabolites differently, then no quantitative differences should be observed between the two conditions. If on the other hand quantification of ^13^C-features results in differing abundances for the same metabolite, then differences in the metabolic composition of ^12^C compounds (see Section 2.1.1 and Section 2.1.2) must be the reason for this deviation.

To test this, the same statistical analysis as used in Section 2.1.2 was carried out for the labeled metabolite signals (Figure 5). The score plot of the principal component analysis (PCA) shows that the two sample types overlap, and no clear separation of the samples is achieved (Figure 5a). Similarly, a univariate comparison of labeled metabolite abundances showed that the majority of metabolites were not significantly different (*p* > 0.05 and fold-change < 0.5 or > 2) when paired with control or toxin-treated matrix (Figure 5b). Only 7 metabolites were significantly more abundant. Moreover, on average, a minor relative increase of around 13% towards higher abundances for those in the toxin matrix was observed. Based on these results, it can be concluded that there was low matrix effect when using unsupervised statistical methods. This is in contrast to an earlier study by Bueschl and Kluger et al. [1]. There, the investigated samples of a Fusarium wildtype and knock-out strain strongly differed in the qualitative metabolic composition (2/3 of the metabolites were not shared between the wildtype and the knock-out strain), which was reflected in a clear separation of samples in the PCA score plot caused by large differences in the observed matrix effects [1].

It shall be noted however that the effect of different matrices on comparative quantification and subsequent statistical analysis has to be evaluated from case to case and might be severe, especially if the metabolic composition is largely altered by experimental perturbations or if different tissue types are compared in biological studies. Thus, the proper evaluation of ion suppression remains a big challenge in untargeted metabolomics studies. Without globally stable isotope-enriched biological samples, the metabolome-wide compensation of potential matrix effects is not possible.

## 3. Materials and Methods

### 3.1. Chemicals

Methanol (MeOH, LC-MS CHROMASOLV®), acetonitrile (ACN, LC-MS CHROMASOLV®), and formic acid (FA, MS grade, approximately 98% purity) were purchased from Riedel-de Haën, Honeywell (Seelze, Germany). The ultrapure water was obtained from an ELGA Purelab system Veolia Water (Ultra AN MK2, Vienna, Austria). ^13^CO_2_ (99% purity) was purchased from Eurisotop (St-Aubin, France), and synthetic air was obtained from Messer (Gumpoldskirchen, Austria). Deoxynivalenol (DON) was purified from *Fusarium graminearum* cultivars and kindly provided by the Institute of Biotechnology in Plant Production (University of Natural Resources and Life Sciences, Vienna (BOKU), Department of Agrobiotechnology (IFA-Tulln), Tulln, Austria).

### 3.2. Plant Material

The uniformly ^13^C-labeled wheat (*Triticum durum*) was produced in the plant cultivation chamber termed labelbox (PhytolabelBox, ECH, Halle, Germany). The same wheat cultivar was grown in the glasshouse (native, unlabeled wheat) at BOKU, IFA-Tulln.

### 3.3. Plant Cultivation

The uniformly ^13^C-labeled wheat was grown in the labelbox (PhytolabelBox, ECH, Halle, Germany) in hydroponic cultures under controlled atmosphere, nutrient delivery, light intensity, and temperature regime. To produce globally ^13^C-labeled plants, the atmosphere in the labelbox contained ^13^CO_2_ instead of ^12^CO_2_. The complete protocol for wheat cultivation is detailed in [37]. At flowering stage (day 87 after the placement into labelbox), the wheat ears were treated inside the labelbox with the fungal toxin deoxynivalenol (DON) or H_2_O, respectively. These two materials were designated L-toxin and L-control; 144 h after the treatment, the plants were harvested and shock frozen in liquid N_2_. For long storage times, the ears were freeze-dried (Labconco FreeZone 6Plus; Labconco, Kansas City, MO, USA) until 5% rest moisture and stored at −80 °C until further use.

The cultivation of native wheat started with the planting of vernalized wheat seedlings in soil filled pots. The native wheat was grown in a glasshouse under controlled temperature and light regime as described in [40]. Similar to the labeled plants, the native wheat plants were treated with DON and H_2_O at onset of flowering. These two materials are designated as toxin and control; 144 h after treatment, the plants were harvested, shock frozen in liquid N_2_, and stored at −80 °C until further use.

### 3.4. Sample Preparation for LC-HRMS Analysis

The fresh native (toxin and control) and dried labeled (L-toxin and L-control) wheat ears were milled separately under cooled condition for 30 s at 40 Hz to a fine powder with a ball mill (MM400, Retsch, Haan, Germany). The dried L-toxin and L-control wheat powders were rehydrated with H_2_O (70 µl/30 mg dried wheat powder) in frozen condition immediately before extraction. Further, the native (100 mg) and rehydrated labeled (30 mg) wheat ear powders were extracted, each with 1 ml of a mixture of ACN/MeOH/H_2_O (3/3/2) + 0.1% FA according to [48]. The L-control and L-toxin extracts were mixed 1/1 (v/v) in one pool (GLMe-IS). Three biological replicates of control and toxin extracts were spiked with the same volume aliquot of the GLMe-IS (1/1 (v/v); see Figure 1a. Additionally, H_2_O + 0.1% FA was added to achieve 1:1 organic solvent–water ratio in the diluted extracts for LC-HRMS analysis.

### 3.5. LC-HRMS Measurement

The LC-HRMS measurements were performed on the QExactive HF-Orbitrap high-resolution mass spectrometer coupled to the Vanquish UHPLC system (Thermo Fisher Scientific). The following settings were used:

#### 3.5.1. Liquid Chromatography

The samples were maintained at 10 °C in the auto sampler, and the column was thermostated to 25 °C. The reversed-phase separation was performed on a C18 column (3.5 µm; 2.1 × 150 mm, XBridge®, Waters, Milford, MA, USA) coupled to a C18 precolumn (4 × 3 mm, Phenomenex, Torrance, CA, USA). The injection volume was 2 µl. The eluent system consisted of (A) H_2_O + 0.1 % FA and (B) MeOH + 0.1 % FA that was passed through the column at a flow rate of 0.25 ml/min with a chromatographic gradient according to [1]. Here, a linear gradient was utilized. The initial mobile phase composition was 10% of eluent B, which was held constant for 2 min. In the next 30 min, eluent B was increased linearly to 100% and maintained at 100% for another 4 min. Lastly, the column was recalibrated for 8 min with 10% eluent B.

#### 3.5.2. Mass Spectrometry

Full scan MS measurements were performed in positive mode, and resolving power setting 120,000 at m/z 200 and m/z range of 100 to 1000 were chosen. The heated electrospray ionization (HESI) source was operated at 320 °C, and +3.5 kV was applied to the spray needle. The auxiliary and sheet gas flow rates were 5 and 55 units, and the setpoint for the auxiliary gas heater was set to 350 °C.

### 3.6. Data Processing and Statistical Evaluation

The LC-HRMS raw data were converted into the mzXML file format via MSConvert of the ProteoWizard open-source software package [50]. The files were further processed by the in-house developed MetExtract II software according to the procedure described in [8]. Briefly, the software searched for pairs of co-eluting chromatographic peaks, which represent both the native and the uniformly ^13^C-labeled metabolite form. All other LC-HRMS features were discarded. For each such found chromatographic peak pair, the *m/z* value for the monoisotopic form, retention time, charge number, mass ion species and neutral loss (if possible), and its total number of carbon atoms (derived from the difference between the monoisotopic and fully ^13^C-labeled isotopologs) were automatically calculated from the data. Moreover, the software matched ions from different samples into a comprehensive data matrix and integrated the peak areas of the detected monoisotopic and fully ^13^C-labeled isotopologs.

All statistical analyses were implemented in the R programming language for statistical computing [19] (version 3.5.3). Concerning missing value imputation, a hybrid approach of omitting features with missing values and replacing missing values with zero was used. More precisely, in the first step of this approach, features missing in some data files (i.e., missing in 1, 2, … or *n*-1 replicates of any experimental group) were omitted. By this, all features that were detected in all replicates of either experimental group (i.e., control or toxin-stressed) and, at the same time, were missing (i.e., not detected) in all replicate samples of the respectively other group were replaced with 0 in all replicates of the latter. With this strategy, 95 features were removed, while another 72 were set to zero in all replicates of only one experimental group. For multivariate analysis, the data matrix was mean-centered and auto-scaled [51]. Ellipses in the PCA scores plot indicate the 95% confidence interval. For univariate analysis, a non-paired, two-sided t-test was calculated and a critical alpha value of 0.05 was chosen together with a minimum mean-fold-change of ≥2 or ≤0.5 as a significance threshold. Metabolites only detected in the control or the toxin group were set to a fold change of infinite (i.e., only present in the toxin group) or 0 (i.e., only present in the control group).

### 3.7. Quality Control

To test the analytical performance of the instrument during the measurement (i.e., intra-sequence technical variability and carry over), a quality control standard mixture (QC-Std, 0.5 mg/L per compound) and solvent blanks were measured at regular intervals throughout the measurement sequence. The 20 compounds contained in the QC-Std were N-methylanthranilate, ferulic acid, syringic acid, methyl-indole-3-carboxylate, indole-3-acetonitrile, kaempferol, 4-triacetate lactone, l-tryptophan, galangin, 3’,4’,5’-O-trimethyltricetin, orientin, schaftoside, thujopsene, chrysoeriol, l-phenylalanine, deoxynivalenol, feruloylputrescine, coumaroylserotonine, feruloylserotonine, and reserpine and were eluted over a wide range of retention times. Besides QC-Std and blank samples, the sequence consisted of 30 biological samples, 6 of which are presented in this study. Biological samples were randomly distributed in-between QC and blank samples in the sequence. Evaluation of QC-Std chromatograms showed an average relative standard deviation ± 9% with respect to peak areas, while the standard deviation of retention times (RT) was ± 0.01 min. None of the metabolic features used for the statistics were detected in the blank samples, and therefore, no carryover was observed.

## 4. Conclusions

The reliable and efficient assessment of the performance of untargeted LC-HRMS-based metabolomics approaches remains to be a major challenge. The present study demonstrates that stable isotope-assisted techniques can be successfully used to evaluate metabolome coverage as well as the accuracy of comparative quantification of the detected metabolites. A globally ^13^C-labeled ECM biological extract was used to detect all truly biology-derived metabolites as well as to assess and compensate putatively different ion suppression/enhancement effects that may occur in samples representing different experimental conditions. Here, plant treatment with the mycotoxin deoxynivalenol was used to exemplify abiotic stress conditions and to illustrate the induction of numerous treatment-specific compounds. The presented approach can be easily extended to both other biological systems as well as further experimental treatments. Here, a total of 68 metabolites (making up 6% of the total detected metabolome) were uniquely found either in toxin-treated or H_2_O-treated control samples. These metabolites were reliably detected as treatment-related metabolites with the aid of GLMe-IS. Further, such an application may also benefit the subsequent analytical steps aiming at sum formula elucidation and annotation of unknown metabolites. Amongst the metabolites which are shared between both experimental conditions, roughly 26% were found to have significantly different abundances. As those metabolites are assumed to be significantly affected by the toxin treatment, they would be of most interest in a typical biological study. Hence, reliable significance testing is crucial in untargeted studies, as based on this, candidate metabolites would be selected and annotated for biological interpretation and even be purified for full structure characterization as well as further biological studies. While under the conditions applied in the present study, matrix effects (i.e., ion suppression/enhancement) turned out to be only moderate, still a considerable proportion of the metabolites, which were assumed to differ significantly between the two conditions, were wrongly classified. It is important to mention that the effect of different matrices on comparative quantification and subsequent statistical analysis is a potential error source of untargeted metabolomics studies and has to be evaluated from case to case. Wrong classification of metabolites as a result of significance testing might be severe, especially if the metabolic composition is largely altered by experimental perturbations or if different tissue types are compared in biological studies. While the accurate assessment of (absolute) matrix effects remains challenging, GLMe-IS enables the metabolome wide compensation of matrix effects. This can be used for a more accurate comparative quantification, especially in untargeted metabolomics applications.

## Figures and Tables

**Figure 1 metabolites-10-00434-f001:**
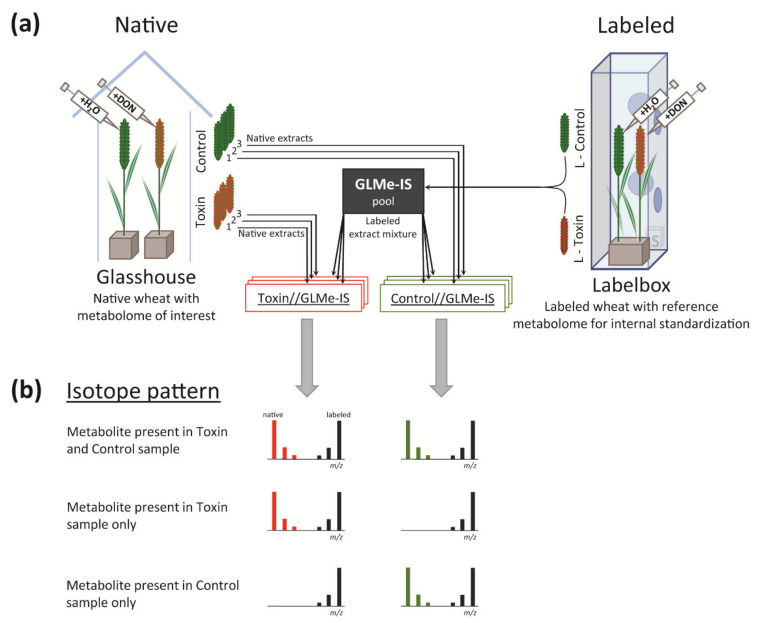
(**a**) Experimental setup to generate native and ^13^C-labeled wheat ear sample extracts: deoxynivalenol (DON) was used for toxin treatment, while water served as the control. (**b**) Isotope pattern in the mass spectrum for three exemplary metabolites: the labeled form is present for all metabolites as it derives from the globally ^13^C-labeled metabolite extract (GLMe-IS) pool, while the presence of native forms may depend on the sample type (i.e., toxin and control).

**Figure 2 metabolites-10-00434-f002:**
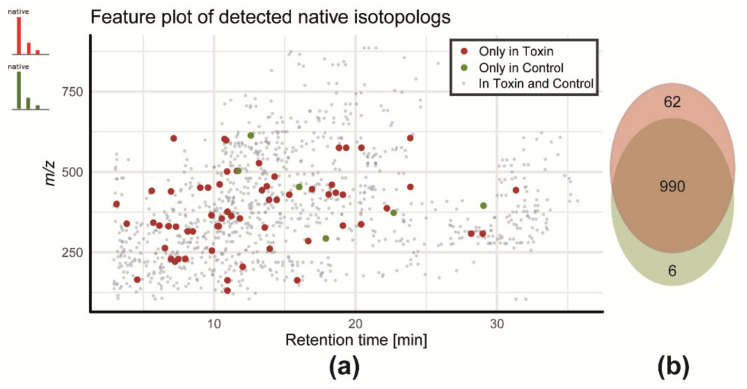
(**a**) Feature plot of all detected native metabolic features and (**b**) qualitative distribution of native metabolites in control- and toxin-treated wheat ear extracts: toxin only (red), control only (green), or common to both sample types (grey). Only those native features that have a labeled analog in GLMe-IS could be detected by MetExtraxt II.

**Figure 3 metabolites-10-00434-f003:**
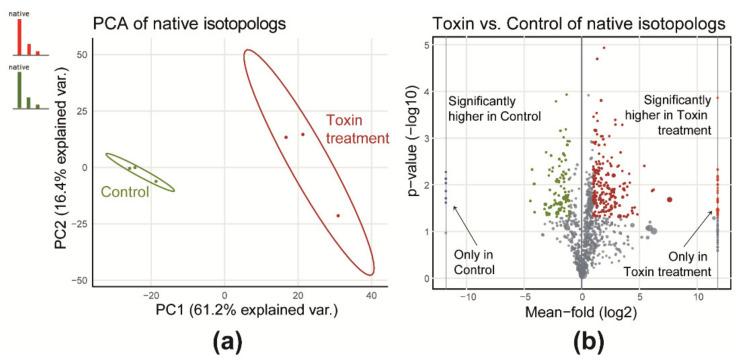
(**a**) Principal component analysis (PCA) scores plot considering all commonly detected ^12^C native metabolic features for control and toxin and (**b**) a volcano plot for a toxin and control comparison of native metabolic features.

**Figure 4 metabolites-10-00434-f004:**
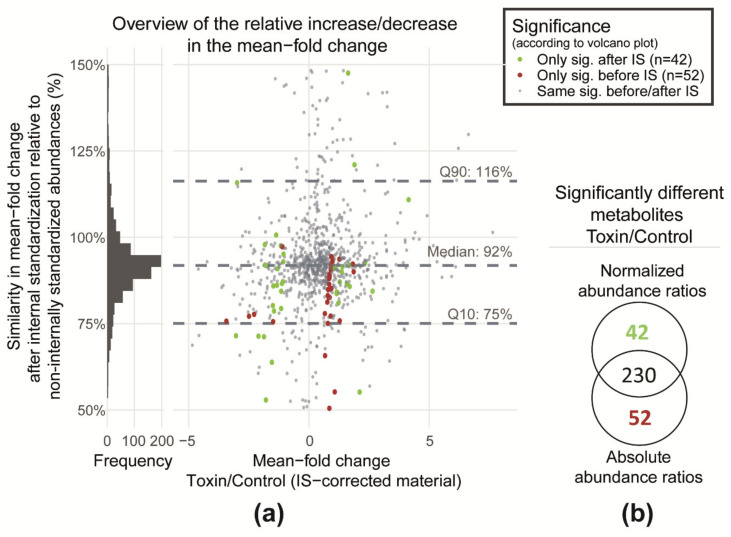
(**a**) Correlation of abundance ratios between control and toxin constituents with and without normalization to ^13^C-metabolites of GLMe-IS (i.e., absolute and normalized abundances) and (**b**) a Venn diagram showing the distribution of significantly differing metabolites of the control and toxin comparison between two groups of the comparative significance test that is obtained with the application of native absolute and normalized abundance ratios.

**Figure 5 metabolites-10-00434-f005:**
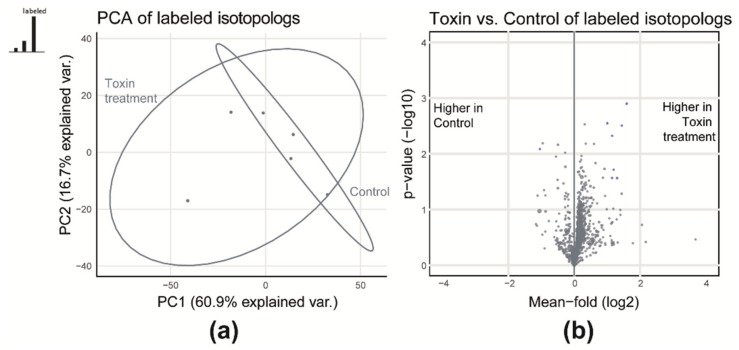
(**a**) PCA and (**b**) volcano plot to demonstrate that the native matrices have similar effects on absolute abundances of labeled metabolites in both sample types.

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
