# Peer review of "Enhanced Metabolome Coverage and Evaluation of Matrix Effects by the Use of Experimental-Condition-Matched 13C-Labeled Biological Samples in Isotope-Assisted LC-HRMS Metabolomics"

_metabolites, 2020, doi:10.3390/metabo10110434_

Round 1

Reviewer 1 Report

In this manuscript, the authors demonstrated the use of stable isotope-assisted approaches to overcome the matrix effect in LC-MS based metabolomics. Overall, the experiment is well-designed, and the results support the conclusion. I have some comments below:

(1) The authors may extend their experiment to negative mode scan and use HILIC column to detect hydrophilic metabolites in the future;

(2) I am little bit confused about " enhance the metabolome coverage". Compared with unlabeled approach, this approach add GLMe-IS as internal reference. However, this would only overcome the ion suprression effect but would not increase the MS sensitivity. The unlabeled approach may also detect many metabolites features, though some of them are not real metabolites, maybe "increase the confidence level" is a better way to express?

Author Response

Response to Reviewer1's  Comments

Note: Please consider that the changes in the manuscript are cited in the Response point: lines correspond to the location of the change in the revised manuscript

Point 1: The authors may extend their experiment to negative mode scan and use HILIC column to detect hydrophilic metabolites in the future;

Response 1: Thank you for the suggestion. We already have some experience with data processing in pos/neg switching mode. We will take this into account and do also plan to extend the LC separation to HILIC, which will certainly further complement metabolite coverage.

Point 2: I am little bit confused about " enhance the metabolome coverage". Compared with unlabeled approach, this approach add GLMe-IS as internal reference. However, this would only overcome the ion suprression effect but would not increase the MS sensitivity. The unlabeled approach may also detect many metabolites features, though some of them are not real metabolites, maybe "increase the confidence level" is a better way to express?

Response 2: We acknowledge this comment. The expression “enhanced metabolome coverage” is meant to describe the advantage of experimental-condition matched internal standard compared to non-experimental-condition-matched material. We agree that this may be a bit misleading in the context of isotope-assisted versus conventional approaches. We have therefore tried to make this clearer and have revised the manuscript in the following way:

(Abstract, line 22): “Here, we demonstrate at the example of chemically stressed wheat that metabolome-wide internal standardization by globally 13C-labeled metabolite extract (GLMe-IS) of experimental-condition-matched biological samples can help to improve the detection of treatment-relevant metabolites and aid in the post-acquisition assessment of putative matrix effects in samples obtained upon different treatments.”

(Introduction, line 155-156): “The ECM GLMe-IS has potential to improve untargeted metabolomics approaches by increasing the confidence level for the detection of truly biology derived metabolites and by compensating putatively different matrix effects between samples obtained upon different treatments.”

Reviewer 2 Report

The authors forgot to use the explanation for the abbreviations the first time they appear in the abstract and also in main text (e.g. LC-HRMS, NMR, GC-MS). 

Line 35 The authors could use also the other keyword (e.g. GLMe-IS)

Author Response

Note: Please consider that the changes in the manuscript are cited in the Response point: lines correspond to the location of the change in the revised manuscript.

Point 1: The authors forgot to use the explanation for the abbreviations the first time they appear in the abstract and also in main text (e.g. LC-HRMS, NMR, GC-MS). 

Response 1: Thank you for pointing this out. The abbreviations are included in the abstract and the introduction sections of the revised manuscript. LC-HRMS is abbreviated with liquid chromatography – high resolution mass spectrometry; GC-MS with gas chromatography – mass spectrometry; and NMR with nuclear magnetic resonance.

(Abstract, lines 18 – 19): “Stable isotope-assisted approaches can improve untargeted liquid chromatography – high resolution mass spectrometry (LC-HRMS) metabolomics studies.”;

(Introduction, lines 39-41): “Alongside with nuclear magnetic resonance (NMR) and gas chromatography – mass spectrometry (GC-MS), the most dominant technique in metabolomics studies is liquid chromatography – high resolution mass spectrometry (LC-HRMS)”.

Point 2: Line 35 The authors could use also the other keyword (e.g. GLMe-IS)

Response 2: Thanks. GLMe-IS has been added to the list of keywords

(Keywords, line 36): “Untargeted metabolomics, internal standard, Deoxynivalenol, abiotic stress of wheat, matrix effects, GLMe-IS”

Reviewer 3 Report

This manuscript describes the use of a globally labeled biological metabolite extract to assess matrix effects and apply broad internal standardization. I appreciate the content and subject of this work. Matrix effects are a very important factor in metabolomics data quality, and efforts to find ways to correct for them and improve data analysis are of significant benefit to the field.

Specific Comments:

1. The authors frequently use the term "metabolites" to refer to detected features in the dataset, both identified and unidentified. I hesitate making claims such as "we detected X metabolites significantly elevated in Group A" when most of them are unidentified. As the authors describe in the introduction, it is well known that most peaks in a metabolomics dataset are artifacts, fragments, contaminants, noise, etc., so please consider referring to unidentified species as "features" or something similar rather than "metabolites". 

2. Paragraph 3 of the Introduction, Line 74 - co-eluting metabolites influencing ionization is only one example of a cause of matrix effects. There are other inherent properties of a sample that cause matrix effects rather than co-eluting metabolites (salt content, etc.). Please provide a more comprehensive description of the nature of matrix effects.

3. Please add references to Introduction paragraph 4 beginning with "Consequently..." (Line 86). These are important points that should be supported by the literature.

4. The Introduction should include a section dedicated to describing current global labeling approaches that are already used in the field (IROA, Credentialing, etc.) and how the approach in this study compares to what has already been done. The use of globally labeled metabolite extracts for internal standardization, metabolite identification, normalization, feature reduction, etc. is not entirely new, so it is important to have this section to be transparent in terms of giving credit to preceding work.

5. Consider abbreviating "experimental-condition-matched". This is a long term frequently used in the text.

6. Line 145 - What is your RT reference?

7. Line 153 - "In total, 1080 metabolites were detected..." - See Comment #1. How do you know these are metabolites? Consider using another word for all descriptions of general peak detection, such as "features". The word "metabolite" is typically properly used when referring to a known species that has been identified as a metabolite.

8. Section 2.1.1 - While the authors state that the purpose of the work was not to conduct a detailed annotation or identification of detected metabolites, information needs to be reported regarding how you identified the metabolites that are specifically discussed. How did you identify these secondary metabolites? What are your parameters? The reader should be convinced that these metabolites are what they are claimed to be.

9. Section 3.5.1 - Please describe the gradient in detail in the text as opposed to referencing it. This information is critical to reproducing the work and is important enough to be included in the manuscript itself.

10. Section 3.5.1 - What was your injection volume?

11. Section 3.5.1 - Did you include blanks, QCs, etc. in your run? How were they used?

12. Line 388 - "Missing values were replaced with 0" - Typically, stats in metabolomics involved replacing missing values with a small value, such as a fraction of the minimum value for a feature, to avoid zeros since zeros can skew statistics. While replacing missing values with 0 may not be incorrect, I would appreciate an explanation for why this was done as opposed to small value replacement.

13. Line 411 - Typo. Might want to do a quick scan for typos throughout the text.

14. No mass spectra are shown. Consider presenting representative mass spectra of your groups as they relate to your assessment of matrix effects. Supplement would be fine.
